# OpenReview forum: "Robust Multi-Objective Controlled Decoding of Large Language Models"
_ICLR.cc/2026/Conference — ICLR 2026 Poster_

### Official Review · Reviewer_uMCr · 2025-10-30

**Soundness:** 3
**Presentation:** 3
**Contribution:** 1
**Rating:** 2
**Confidence:** 3

**Summary:**

The paper *“Robust Multi-Objective Controlled Decoding of Large Language Models”* extends the multi-objective decoding framework by introducing a robustness criterion. Instead of optimizing for a fixed set of user-specified objective weights, the proposed approach seeks a **decoding policy that remains optimal under the worst-case combination of weights**. This effectively ensures robustness to uncertainty or variability in user preferences at inference time. The formulation begins as a bilevel optimization problem, which is then relaxed into a single-level equivalent and further simplified into a closed-form solution, yielding a practical inference-time algorithm (tRMOD) that can efficiently generate robust aligned outputs without retraining.

**Strengths:**

- **Comprehensive experiments:** The evaluation spans multiple datasets and baselines, demonstrating consistent improvements in worst-case rewards relative to previous approaches.
- **Clear theoretical pathway:** The derivation from the bilevel to single-level formulation, and eventually to a closed-form expression, is well presented and technically sound.
- **Practicality:** The reported latency and computational results show that the proposed RMOD algorithm is feasible for real-time inference, making it suitable for deployment in alignment-sensitive applications.

**Weaknesses:**

**Philosophical concern about the problem setup:** While the mathematical formulation is solid, the motivation for a *universal robust inference-time policy* is debatable. Inference-time policies are typically **user-specific**, reflecting individual objective preferences. Each user can specify new weights or objectives, and the decoding process adapts accordingly. Designing a single robust policy to handle all users’ worst-case preferences might conflict with the personalized spirit of inference-time alignment.

The paper would benefit from explicitly clarifying this conceptual distinction relative to *SitAlign* (Chehade et al., 2025), which also explores user-conditioned value weighting at inference time.

The notion of “worst-case” robustness could be better motivated—what kind of real-world variability in user preferences is being modeled, and why is the minimax setup the most appropriate solution?


### Reference

- Chehade, Mohamad, et al. *"Bounded Rationality for LLMs: Satisficing Alignment at Inference-Time."* arXiv preprint arXiv:2505.23729 (2025).

**Questions:**

1. How do you justify the need for a single robust policy that optimizes for the worst-case user preference, instead of learning user-conditioned inference-time policies as in SitAlign?
2. What practical scenarios justify assuming adversarial or worst-case user weights, given that preferences can typically be queried or provided interactively?
3. How sensitive is tRMOD’s performance to the assumed weight uncertainty set? Could over-conservatism lead to underperformance in typical (non-worst-case) users?
4. Can this framework be extended to dynamically update the robust policy as user preferences change, rather than precomputing a single static one?

---

> ### Author Response · Authors · 2025-11-19
>
> Dear Reviewer uMCr,
>
> Thank you for your review and your acknowledgement of our paper’s comprehensive experiments, theoretical clarity, and practicality. We also thank the reviewer for pointing out RMOD’s potential relation to SitAlign. We have cited the paper in the updated manuscript (Section 2) to further clarify our work’s contribution. We provide answers to your comments on the setting of our paper below.
>
> > “a single robust policy to handle all users’ worst-case preferences might conflict with the personalized spirit of inference-time alignment.”
>
> A1. As mentioned in the Introduction (line 48-50) and Related Work (second paragraph, Section 2), we consider a setting **without user-specific information**. Importantly, **RMOD is not meant to replace personalized inference-time policies but to serve as a safe and principled default policy when user preferences are unknown, conflicting, or unavailable**. Practical examples of such cases include:
>
> 1. **Privacy:** when a user doesn’t want prior interactions with a model to be stored and used for personalization.
> 2. **First-time interactions:** In a competitive market, chat services might drive users away by immediately asking for personal preferences without first proving they are useful.
>
> RMOD uses explicit weights over objectives allowing later personalization when user data is available and personalization is necessary or wanted by a user (see answer A7). However this is direction for future work and not the specific focus of our paper.
>
> We will clarify these use cases clearly in the introduction given more space in a camera ready version of the work.
>
> > “How do you justify the need for a single robust policy that optimizes for the worst-case user preference, instead of learning user-conditioned inference-time policies as in SitAlign?”
>
> A2. As detailed in A1, there are many scenarios where user information is unavailable or impractical to gather and a robust policy provides a strong initialization in these settings.
>
> > “The paper would benefit from explicitly clarifying this conceptual distinction relative to SitAlign (Chehade et al., 2025)”
>
> A3. The difference between the settings of RMOD and SitAlign is the **availability of user-specific information**. RMOD does not require any of this information at test-time, while SitAlign assumes access to personalized threshold information on the objectives which require LLM evaluation or human feedback (last paragraph of Section 3.1 in Chehade et al., 2025)
>
> > “The notion of “worst-case” robustness could be better motivated, … why is the minimax setup the most appropriate”
>
> A4. In A1 we provide examples of settings where personalization isn’t desirable. In these cases we might often prefer a policy that minimizes user dissatisfaction. We often care about competing objectives e.g. helpfulness vs. harmlessness, or conciseness vs. informativeness (see Intro paragraph 2), solving a maxmin optimization problem finds a solution that **optimally balances these competing objectives**. We see this in Figure 3b) where uniform weights across the HH objectives result in responses that prefer being helpful over harmless, RMOD balances these competing objectives and narrow the gap between them.
>
> > “How sensitive is tRMOD’s performance to the assumed weight uncertainty set?”
>
> A5. As RMOD does not assume any information about the weights, it considers the set of all possible weight vectors for computing $\hat{w}^\*$.  As long as the uncertainty set allows gradient descent to find $\hat{w}^\*$, RMOD’s performance will not be sensitive with respect to the uncertainty set.
>
> > “Could over-conservatism lead to underperformance in typical (non-worst-case) users?”
>
> A6. By changing the value of the hyperparameter $\lambda$, RMOD can control its level of robustness or conservatism to avoid underperformance in non-worst-cases. Figure 3-(c) demonstrates this, where RMOD concentrates further to the worst-case objective as the value of $\lambda$ increases. When $\lambda$ is set to an appropriate value such as 0.5 in Figure 3-(b), RMOD achieves both robustness and reasonable performance improvement in all the objectives.
>
> > “Can this framework be extended to dynamically update the robust policy as user preferences change?”
>
> A7. We agree with the reviewer that this is an interesting idea and we look forward to addressing these classes of problems in our future work. As RMOD places explicit weights across objectives, it is possible to combine it with other algorithms and transition from setting these weights in a robust manner to a personalized one.
>
> If you have any further unresolved concerns or questions regarding this paper, we would be more than happy to address them. If all concerns have been resolved, we would kindly ask you to raise the score of this paper in light of the responses provided above.
>
> Best regards,
>
> The authors

---

> > ### Comment · Reviewer_uMCr · 2025-11-21
> >
> > I thank the authors for their wonderful addressing of my concerns. I still have one left, though. If we don't know the user preferences, why do we have to wait until inference time to modify the policy responses? Why can't we do that during the RLHF?

---

> > > ### Author Response · Authors · 2025-11-21
> > >
> > > [2nd Response]
> > >
> > > Dear Reviewer uMCr,
> > >
> > > Thanks for your response. We are happy to address your remaining concern. We provide three main reasons why the inference-time adaptation by RMOD can be preferred over robust RLHF:
> > >
> > > 1. **There are cases where we cannot fine-tune a base LLM**. Inference time methods such as Controlled Decoding do not assume access to the model weights, or the ability to train a model (Introduction paragraph 1 of [1]). Approaches like RMOD have a wider applicability than training time methods, particularly when the reference policy weights are not accessible or when the parameter size is very large.
> > > 2. **RMOD performs better than RLHF-based training time methods**, in terms of worst-case performance. This was demonstrated in our experimental comparison to Maxmin-GRPO (line 989-1012 of the updated manuscript), shown in Figure 2. We hypothesise that test-time exploration leveraged by RMOD resulted in this advantage. We implement our own fine-tuning baseline because the existing method [3] requires additional user group information.
> > > 3. **RMOD is more flexible than training time approaches.** RMOD can easily adapt to the change in the objective set. RMOD can be run on any combination of the objectives without retraining and can even be composed with an additional value function.
> > >
> > > We’d like to thank you for taking the time to respond to our rebuttal, and we hope we have addressed your concern. If you have any further concerns, please do let us know; otherwise we would kindly ask you to consider raising your score.
> > >
> > > Kind regards,
> > >
> > > The Authors
> > >
> > > [1] Mudgal, Sidharth, et al. "Controlled decoding from language models." arXiv preprint arXiv:2310.17022 (2023).
> > >
> > > [2] Chehade, Mohamad, et al. "Bounded Rationality for LLMs: Satisficing Alignment at Inference-Time." arXiv preprint arXiv:2505.23729 (2025).
> > >
> > > [3] Chakraborty, Souradip, et al. “Maxmin-rlhf: Towards equitable alignment of large language models with diverse human preferences”. arXiv preprint arXiv:2402.08925 (2024).

---

> > > > ### Comment · Reviewer_uMCr · 2025-11-22
> > > >
> > > > My concerns have been addressed. I shall raise my score.

---

> ### Author Response · Authors · 2025-11-24
> **Appreciation for Reassessing the Paper**
>
> Dear Reviewer uMCr,
>
> Thank you very much for your active engagement in the discussion and for kindly reassessing our work. We are glad that our exchange helped clarify the contributions of our paper, and we truly appreciate your careful consideration.
>
> Best regards,
>
> The authors

---

### Official Review · Reviewer_FxTQ · 2025-10-31

**Soundness:** 3
**Presentation:** 3
**Contribution:** 2
**Rating:** 4
**Confidence:** 4

**Summary:**

The paper focuses on multi-objective controlled decoding to perform robust inference-time alignment across multiple objectives. A key challenge with multiple, often conflicting, objectives is determining how to balance them effectively. To address this, the authors propose a maximin-style approach that maximizes the worst-case reward combination rather than relying on fixed or pre-learned weights. They formulate the decoding objective as a max–min problem and demonstrate a clean convex structure in the objective. The paper further introduces an efficient decoding algorithm that leverages value functions trained per objective, iteratively updating the weights during decoding to achieve robust alignment without requiring further retraining. Empirical results are promising across the alignment datasets.

**Strengths:**

- The paper is a most natural extension to the Maxmin RLHF training time approaches which is interesting.
- The observation of convexity of the objective due to the Logsumexp function is interesting (although similar has been leveraged in certain past works)
- Multiple value functions are trained in CD-Fudge style with objective-specific rewards using a reference model which is crucial.
- Experimental results are clean and shows gains with multiple objectives.
- The observation and experimental setting by increasing the difficulty and increasing number of objective to worst case reward is particularly interesting

**Weaknesses:**

- One of the key weakness lies in the way how the value function are trained. The value function are trained with data from reference policy which means its restricted to V^pi and not V^pi* which will result in sub-optimality (check Transfer Q*, Bounded Rationality for LLMs: Satisficing Alignment at Inference-Time  for reference) otherwise needs to meet certain coverage assumptions on the policy.]
- Whats the data on which the value functions are trained? They have been generated under which policy? Its extremely crucial to understand.
- How are the value functions trained? Do they use the linear head with the same backbone? Since, for conflicting objective these structure can have some issues?
- Experimental comparisons with training time Maxmin objectives needs to be performed and highlighted why the gain is coming? or if not why it performs better. Due to Test-time exploration, inference objectives can do better than training also (Transfer Q^*). Will be helpful to provide details.

One important thing is to provide a detailed discussion of the value function training, challenges, uniqueness in this setting of multiple policies and coverage aspects.

**Questions:**

Check in Weakness section

---

> ### Author Response · Authors · 2025-11-19
>
> Dear Reviewer FxTQ,
>
> We’d like to thank the reviewer for recognising the interesting theoretical analysis of our objective and the clean and extensive nature of our experiments and ablations. We also thank the reviewer for pointing out RMOD’s potential relation to TQ* and SitAlign. We have cited these papers in the updated manuscript (Section 2) to further clarify our work’s contribution. To address the reviewer’s concerns and questions:
>
> > “Provide extensive details on the data, training, and architecture of the value functions”
>
> ### Data
>
> A1.  As described in the Datasets paragraph of Section 6.1, line 349, we generate 4-16 trajectories per prompt from the reference policy as a training dataset.
>
> ### Training and Architecture
>
> A2. The details of value function training are covered in Appendix C.1, due to space constraints. We train our value functions using the CD-FUDGE objective shown in Eq. 8. As specified in Appendix C.1, we used an LLM model as a shared backbone and have a linear head per objective. We used this architecture as it reduced the memory requirements of RMOD, a key practical consideration as our method requires both a reference model and a value model to be loaded at inference time. As seen in Figures 3 b) & c), this architecture can guide the reference model across a range of different helpful and harmless behaviours, despite the conflicting nature of these objectives.
>
> > “Value functions are restricted to $\pi_\mathrm{ref}$ and a coverage assumption is needed for optimality”
>
> A3. We thank the reviewer for raising this point. Our method, like Controlled Decoding, trains a **value function only for the reference policy $\pi_\mathrm{ref}$, not for the optimal policy $\pi^*$**. Because of this, **we do not require global coverage assumptions needed for off-policy evaluation or optimal value estimation as in [1] or [2]**. To estimate $V^{\pi_\mathrm{ref}}$, it is sufficient that trajectories from the reference policy cover its own trajectory distribution, which is the standard assumption in CD and our work. RMOD is designed to improve robustness of CD conditional on an accurate value function, and its guarantees do not rely on $\pi_\mathrm{ref}$ being optimal nor on estimating $V^{\pi^*}$. Thus, issues of policy coverage or optimality are orthogonal to the robustness improvement RMOD provides. Our experiment in Figure 2 demonstrates that value functions trained with CD-FUDGE achieve better alignment performance than MO-GRPO, DPO, RS, and MOD, supporting the empirical soundness of our approach.
>
> > Experimental comparisons with fine-tuning baselines with Maxmin Objectives
>
> A4. Thank you for this suggestion. As prior works on robustness i.e. Maxmin-RLHF [3], require user subpopulation data, these works aren’t directly relatable to ours. However, we instead tested a robust multi-objective variant of GRPO (Maxmin-GRPO) - full details are provided in Appendix C.1 of our updated manuscript. The results on the HH dataset are as follows:
>
> | Algorithm | Harmless reward | Helpfulness reward | Worst-case reward |
> | ----------- | ----------------------------------------- | ---------------------- | ------------------ |
> | **RMOD** | 1.685 $\pm$ 0.026 | 2.016 $\pm$ 0.035 | **1.270 $\pm$ 0.027** |
> | RMOD-distill | 1.447 $\pm$ 0.03  | 1.994 $\pm$ 0.038 | 1.005 $\pm$ 0.029 |
> | MO-GRPO | 1.018 $\pm$ 0.035 | 2.267 $\pm$ 0.04 | 0.782 $\pm$ 0.034 |
> | **Maxmin-GRPO** | 1.004 $\pm$ 0.034 | 1.854 $\pm$ 0.04 | 0.604 $\pm$ 0.031 |
> | GRPO-harmless | 1.153 $\pm$ 0.034 | 1.374 $\pm$ 0.045 | 0.449 $\pm$ 0.034 |
> | GRPO-helpful | 0.332 $\pm$ 0.032 | 3.225 $\pm$ 0.467 | 0.265 $\pm$ 0.031 |
>
> Maxmin-GRPO fails to improve the worst-case reward beyond that of MO-GRPO despite being trained on a robust objective. We hypothesise that the minimum operation over the objectives in Eq. 33 (See Appendix C.1 of the updated manuscript) removes the training signal of other objectives that are not the worst-case. RMOD does not suffer from this issue and utilizes all the objectives when steering the response generation.
>
> We’d once again like to thank the reviewer for their time. In light of our comments, updated manuscript, and additional experiments, we ask that they raise the score of our work.
>
> Best regards,
>
> The authors
>
> [1] Chehade, Mohamad, et al. "Bounded Rationality for LLMs: Satisficing Alignment at Inference-Time." arXiv preprint arXiv:2505.23729 (2025).
>
> [2] Chakraborty, Souradip, et al. "Transfer q-star: Principled decoding for llm alignment." Advances in Neural Information Processing Systems 37 (2024): 101725-101761.
>
> [3] Chakraborty, Souradip, et al. “Maxmin-rlhf: Towards equitable alignment of large language models with diverse human preferences”. arXiv preprint arXiv:2402.08925 (2024).

---

> > ### Comment · Reviewer_FxTQ · 2025-11-28
> > **Response to rebuttal by the authors**
> >
> > I thank the authors for their detailed response which helps in clearing some of my concerns.
> >
> > However, i am still confused about the fact that the authors claim they dont require any coverage assumptions. Can the author please provide an explanation why estimating V_pi_ref estimating is sufficient for controlled decoding. To the best of my understanding in the CD paper, it was an approximation as highlighted in Transfer-Q*. I wanted to understand the impact in this scenario?
> >
> > "We hypothesise that the minimum operation over the objectives in Eq. 33 (See Appendix C.1 of the updated manuscript) removes the training signal of other objectives"
> > I dont' understand this statement clearly, since the min is dependent on V_pi(group) and not for a specific group so as policy changes, the minority group changes as well as its a function of policy. Hence, the results are not clear. Can the authors clearly explain the improvement.
> >
> > I thank the authors again for their additional ablation and studies.

---

> > > ### Author Response · Authors · 2025-12-02
> > >
> > > Dear Reviewer FxTQ,
> > >
> > > Thank you for your additional comments in response to our rebuttal.
> > >
> > > > “Can the author please provide an explanation why estimating V_pi_ref is sufficient for controlled decoding. To the best of my understanding in the CD paper, it was an approximation as highlighted in Transfer-Q*.”
> > >
> > > We thank the reviewer for the follow-up question. Using only $V_{\pi_\mathrm{ref}}$ for CD is sufficient because it **focuses on improving the quality of the generation compared to the reference policy.** As CD does not rely on the concept of optimal value function, it does not require a coverage assumption. We chose CD for our implementation of RMOD, because Transfer-Q* has a **significant trade-off in its practical implementation, requiring generating a complete sequence for the top-k tokens at each decoding step (line 7, 9 of Algorithm 1 in [1]).** Instead, using a trained value function enables quicker generation, whose empirical performance is well demonstrated in the experiments of our paper.
> > >
> > > > “min is dependent on V_pi(group) and not for a specific group so as policy changes, the minority group changes as well as its a function of policy” “Can the authors clearly explain the improvement.”
> > >
> > > This comment refers to the Maxmin-GRPO, a training time algorithm that we have run additional experiments on. In this setting **there are no additional $V_{\pi,g}$**. Instead, at each training instance, **rewards of a single objective are selected, $r_{g,min}$ as detailed in Eq.34 and Eq.35 in our paper**, and are used for training. As the algorithm selects only a single objective, it does not receive information about the other objectives. We compare this to Multi-Objective (MO) GRPO, which does receive this additional information. We hypothesise that MO-GRPO outperforms Maxmin-GRPO in terms of the worst-case reward despite MaxMin-GRPO directly optimizing for this, because MO-GRPO has access to all the objectives during training, whereas MaxMin-GRPO does not. RMOD outperforms all the GRPO baselines due to the advantage of test time exploration, **using the value functions of all the objectives to compute $\hat{w}^*$ and select blocks.**
> > >
> > > We hope the answers above have provided enough clarification to your questions, and kindly ask for your reassessment of our paper in light of this response.
> > >
> > > Kind regards,
> > >
> > > The authors
> > >
> > > [1] Chakraborty, Souradip, et al. "Transfer q-star: Principled decoding for llm alignment." Advances in Neural Information Processing Systems 37 (2024): 101725-101761.

---

> ### Author Response · Authors · 2025-11-24
>
> Dear Reviewer FxTQ,
>
> We would like to gently remind you that we have submitted a response to your initial review. We would greatly appreciate it if you could let us know whether our clarifications address your concerns or if any additional points would be helpful.
>
> Best regards,
>
> The authors

---

### Official Review · Reviewer_GQqL · 2025-11-01

**Soundness:** 2
**Presentation:** 3
**Contribution:** 3
**Rating:** 6
**Confidence:** 2

**Summary:**

In this paper, the authors introduce Robust Multi-Objective Decoding (RMOD), an inference-time algorithm to align LLMs with multiple, competing objectives like helpfulness and safety. RMOD addresses the need to balance these objectives without manual tuning. To be specific, RMOD formalizes the problem as a maximin two-player game between the sampling policy and the objective weights, with the goal of maximizing the reward of the worst-performing objective. To support their hypothesis, the authors theoretically show that this maximin game reduces to a convex optimization problem over the weights, which can be solved to find the optimal weights. The optimal sampling policy is then analytically derived from these weights. Experiments on the HH, UltraFeedback, and ValuePrism datasets show that RMOD achieves a higher worst-case reward and win rate than baselines.

**Strengths:**

1. The paper is well-written in general. The research problem of multi-reward alignment has been well-articulated to the reader. The proposed RMOD algorithm is based on a maximin formulation, which is a principled and well-motivated approach to this problem

2. The theoretical analysis of formulating the max-min two-player game and its reduction to a single, convex optimization problem (Eq. 7) is interesting.

3. Across three standard benchmarks: Helpfulness-Harmless, UltraFeedback, and ValuePrism, RMOD outperforms all the compared baselines (MOD, MO-DPO, CD).

**Weaknesses:**

1. The primary concern lies in the assumption that value functions are available for all objectives. Although Section 5.1 outlines the loss function used to train these value models, this requirement somewhat undermines the inference-time nature of the framework, as it necessitates additional training to obtain the value functions.

**Questions:**

Please refer weaknesses

---

> ### Author Response · Authors · 2025-11-19
>
> Dear Reviewer GQqL,
>
> We appreciate your time and effort in reviewing our work. We are glad that you found the paper well written and the proposed algorithm principled and well motivated.
>
> > The assumption that value functions are available for all objectives undermines the inference-time nature of the framework.
>
> A. Training of value functions is required for all value-based decoding methods, including CD [1], and is not a disadvantage introduced by RMOD. While value function training imposes overhead over Best-of-N, value functions offer several advantages over alignment methods based on policy fine-tuning, particularly in terms of inference-time adaptability.
>
> 1. Different combinations of value functions can be employed depending on the context or the user's choice.
> 2. The strength of alignment can be adjusted without retraining.
> 3. A value function can be reused for a different reference model [1].
> 4. A value function can be smaller than the base model [1, 2, 3].
>
> Should you have any further unresolved concerns or questions regarding this paper, we would be more than happy to address them. If we have addressed all your concerns, we ask that you consider raising your score.
>
>
> Best regards,
>
> The authors
>
>
> [1] Mudgal, Sidharth, et al. "Controlled Decoding from Language Models." International Conference on Machine Learning. PMLR, 2024.
>
> [2] Zhou, Zhanhui, et al. "Weak-to-strong search: Align large language models via searching over small language models." Advances in Neural Information Processing Systems 37 (2024): 4819-4851.
>
> [3] Chakraborty, Souradip, et al. "Transfer q-star: Principled decoding for llm alignment." Advances in Neural Information Processing Systems 37 (2024): 101725-101761.

---

> ### Author Response · Authors · 2025-11-24
>
> Dear Reviewer GQqL,
>
> We would like to gently remind you that we have submitted a response to your initial review. We would greatly appreciate it if you could let us know whether our clarifications address your concerns or if any additional points would be helpful.
>
> Best regards,
>
> The authors

---

### Official Review · Reviewer_k3rX · 2025-11-14

**Soundness:** 3
**Presentation:** 3
**Contribution:** 2
**Rating:** 4
**Confidence:** 3

**Summary:**

This paper proposes Robust Multi-Objective Decoding (RMOD), an inference-time alignment method that aims to robustly satisfy multiple reward objectives. RMOD formulates decoding as a max–min game between a sampling policy and an adversarial distribution over objective weights, and shows this game can be reduced to a convex optimization over the weight simplex with a closed-form best-response policy. For practical use, the authors design a blockwise controlled-decoding algorithm that approximates the equilibrium using K samples from a reference model and iteratively updated weights, plus a distillation scheme that trains a single policy to imitate RMOD outputs. Experiments on Anthropic HH, UltraFeedback, and ValuePrism show that RMOD improves worst-case reward and worst-case win rate over controlled decoding with fixed weights, Best-of-K, and several fine-tuning baselines.

**Strengths:**

1. RMOD provides a principled max–min formulation of multi-objective decoding and reduces it to a convex optimization over objective weights, giving a clear theoretical underpinning for worst-case alignment at inference time.
2. The blockwise controlled-decoding algorithm is practical: it reuses K samples from a reference model, introduces less than ~4.5% additional latency over standard controlled decoding on HH, and includes a distillation procedure that recovers most robustness benefits with a single-response policy.
3. Experiments on HH, UltraFeedback, and ValuePrism consistently show improved worst-case rewards and worst-case win rates compared to fixed-weight controlled decoding, Best-of-K, and multi-objective fine-tuning baselines (GRPO, DPO, Rewarded Soups), while keeping average performance competitive.

**Weaknesses:**

1. The method relies heavily on pre-trained reward models and learned value functions, but the paper provides little analysis of how calibration or misspecification across objectives affects the claimed robustness, which may limit the interpretability of “worst-case” guarantees.
2. Although related robust alignment and group-robust RLHF methods are discussed, the empirical comparisons are restricted to non-robust baselines (fixed-weight controlled decoding and a few fine-tuning approaches), so it remains unclear how RMOD compares to other robustness-oriented techniques on similar datasets.

**Questions:**

How sensitive is RMOD to the relative scaling and calibration of different reward heads, and have the authors tried alternative normalization schemes or diagnostics to detect when one objective dominates due to miscalibration?

---

> ### Author Response · Authors · 2025-11-19
>
> Dear reviewer k3rX,
>
> We’d like to thank the reviewer for their time and for reviewing our paper. We appreciate their recognition of the clear theoretical underpinning of the objective behind our work, the practicality of our algorithm, and the distilled model. We provide the responses to the comments and questions below.
>
> > How does objective (reward) misspecification affect the robustness optimization in RMOD?
>
> A1. Thanks for bringing up an important point. Indeed, significant objective misspecification may undermine robust optimization by confounding the identification of the worst-case objective. However, RMOD can assign soft weights over the objectives (Eq. (5)) to minimize the risk of fully committing to a single misidentified worst-case objective. The smoothness of the soft weights is controlled by the hyperparameter $\lambda$, as shown in Figure 3 (c). With the soft weighting mechanism, RMOD can operate under a minor degree of reward misspecification, demonstrating consistent performance when evaluated by LLM judges, which are not available at training time (Table 1). We will augment the manuscript with this discussion.
>
> > “How sensitive is RMOD to the relative scaling and calibration of different reward heads, and have the authors tried alternative normalization schemes or diagnostics to detect when one objective dominates due to miscalibration?”
>
> A2. The relative scaling of rewards does affect RMOD’s behavior, and we will clarify this point in the updated manuscript. However, this issue is not severe in multi-objective alignment scenarios, which are the main focus of the paper. A reward model for alignment is trained on a preference dataset using the standard RLHF protocol, naturally yielding comparable scales.
>
> Still, in the Ultrafeedback dataset paragraph of Appendix C.3, we do provide a normalization protocol based on standardizing the reward distribution observed under the reference policy. We believe this can be a reliable normalization method to achieve equitable alignment across multiple objectives. We will elaborate on this protocol in the main text when we have additional space.
>
> > “The empirical comparisons are restricted to non-robust baselines”
>
> A3. Thank you for your comment on this. As prior works on robustness, such as Maxmin-RLHF [1], have different settings, they are not directly comparable to our method. We instead tested a robust multi-objective variant of GRPO (Maxmin-GRPO) - full details are provided in Appendix C.1 of our updated manuscript. The results on the HH dataset are as follows:
>
> | Algorithm | Harmless reward | Helpfulness reward | Worst-case reward |
> | ----------- | ----------------------------------------- | ---------------------- | ------------------ |
> | **RMOD** | 1.685 $\pm$ 0.026 | 2.016 $\pm$ 0.035 | **1.270 $\pm$ 0.027** |
> | RMOD-distill | 1.447 $\pm$ 0.03  | 1.994 $\pm$ 0.038 | 1.005 $\pm$ 0.029 |
> | MO-GRPO | 1.018 $\pm$ 0.035 | 2.267 $\pm$ 0.04 | 0.782 $\pm$ 0.034 |
> | **Maxmin-GRPO** | 1.004 $\pm$ 0.034 | 1.854 $\pm$ 0.04 | 0.604 $\pm$ 0.031 |
> | GRPO-harmless | 1.153 $\pm$ 0.034 | 1.374 $\pm$ 0.045 | 0.449 $\pm$ 0.034 |
> | GRPO-helpful | 0.332 $\pm$ 0.032 | 3.225 $\pm$ 0.467 | 0.265 $\pm$ 0.031 |
>
> Maxmin-GRPO fails to improve the worst-case reward beyond that of MO-GRPO despite being trained on a robust objective. We believe Maxmin-GRPO underperforms MO-GRPO as the minimum operation over the objectives in Eq. (33) of the updated manuscript removes the training signal of objectives that are not the worst-case. RMOD does not suffer from this issue as it utilizes the information of all the objectives from value functions while steering the response generation.
>
> If you have any further unresolved concerns or questions regarding this paper, we would be more than happy to address them.
> If all concerns have been resolved, we would kindly ask you to raise the score of our work.
>
> Best regards,
>
> The authors
>
> [1] Chakraborty, Souradip, et al. “Maxmin-rlhf: Towards equitable alignment of large language models with diverse human preferences”. arXiv preprint arXiv:2402.08925 (2024).

---

> > ### Author Response · Authors · 2025-11-24
> >
> > Dear Reviewer k3rX,
> >
> > We would like to gently remind you that we have submitted a response to your initial review. We would greatly appreciate it if you could let us know whether our clarifications address your concerns or if any additional points would be helpful.
> >
> > Best regards,
> >
> > The authors

---

### Author Response · Authors · 2025-12-02
**Summary of Rebuttal Process for the Area Chair**

Dear AC,

Thank you for taking the time to review our work. We appreciate that the last week has likely caused you a huge amount of inconvenience and additional work. We provide a summary of how the discussion, including our rebuttals so far, has addressed all concerns raised in the initial reviews:

## Reviewer GQqL

The reviewer gave an **initial score of a 6** and questioned RMOD’s assumptions. We further clarified how RMOD’s assumptions are in line with the broader literature [2] and thus reasonable.

## Reviewer uMCr

Through our thorough addressing of all the concerns, **the reviewer had raised their score from a 2 to a 6 before the scores were reverted.** The reviewer expressed many concerns about the motivation of our work, including the practicality of having a single robust policy when inference time methods typically allow for personalisation, why specifically we should use a max-min objective, and asked for comparisons to a prior work called SitAlign [1], among other things.

We addressed the reviewer’s comments by providing realistic settings in which our algorithm, RMOD, is a practical solution. We justified the max-min objective by the need to balance competing desirable properties such as helpfulness and harmlessness, and discussed the differences between RMOD and SitAlign’s problem settings to clarify RMOD’s novelty. The reviewer queried why a robust inference-time algorithm was needed vs. a robust training approach, and we addressed this by referencing prior work and referring to our new empirical results in response to other reviewers’ questions.

## Reviewer FxTQ

With the initial score of a 4, the reviewer requested further details around the exact nature and training of the value functions in our paper. We provided these and highlighted their inclusion in the text and appendix of the original manuscript. The reviewer also requested experimental comparisons with max-min training time objectives, and we ran additional experiments on a novel MaxMin-GRPO algorithm that reflects a training time version of our setting.

Finally, the reviewer sought to understand how estimating $V_{\pi_\mathrm{ref}}$ is sufficient for Controlled Decoding (CD) [2], given the theory highlighted in Transfer-Q*. Had the rebuttal period continued, we would have argued that $V_{\pi_\mathrm{ref}}$ is sufficient, as **CD focuses only on improving the reference policy and not finding the optimal action value function** (Section 2 in [2]). Thus, we do not require coverage assumptions to ensure this optimum is in the policy space of CD policies.

We choose to use CD over Transfer-Q* as **CD is a much more practical algorithm that leverages pre-trained value functions with a quick blockwise approach of response generation.** In contrast, despite having theoretical advantages, Transfer-Q* leverages an expensive per-token approach without explicit value functions (line 7&9 of Algorithm 1 in [3]).

## Reviewer k3rX

With the initial score of a 4, this reviewer asked how RMOD is affected by reward misspecification or miscalibration. We clarified that RMOD can control the concentration of weights by using the hyperparameter $\lambda$ to alleviate the issues of misspecification, and that our empirical results with a separate LLM-as-a-judge have shown that RMOD can operate under a minor degree of reward misspecification (Table 1). We also clarified how reward miscalibration is often not a serious issue, and discussed one of our experiments where we apply normalization to reward signals to improve equitable alignment between the objectives. This reviewer also requested a comparison to baselines that are designed for robustness, and we have provided results with the MaxMin-GRPO algorithm that optimizes a robustness objective during training.

Once again, we thank you for your time. We hope this summary of our rebuttal so far has provided useful information for your assessment.

Best regards,

The authors

[1] Chehade, Mohamad, et al. "Bounded Rationality for LLMs: Satisficing Alignment at Inference-Time." arXiv preprint arXiv:2505.23729 (2025).

[2] Mudgal, Sidharth, et al. "Controlled decoding from language models." arXiv preprint arXiv:2310.17022 (2023).

[3] Chakraborty, Souradip, et al. "Transfer q-star: Principled decoding for llm alignment." Advances in Neural Information Processing Systems 37 (2024): 101725-101761.

---

### Meta-Review · Area_Chair_zom3 · 2026-01-09

**Summary:**

The paper introduces a controlled decoding algorithm for multi-objective alignment for maximizing the worst-case reward. The algorithm formulates the problem as a maxmin game between adversarial objective weights and the sampling policy. To solve this problem, the authors propose using gradient descent to optimize the objective weights. The approach is evaluated across several alignment benchmarks with up to 10 objectives and consistently demonstrates improvements in worst-case reward.

Strengths

- A principled max–min formulation of multi-objective decoding, supported by theoretical analysis of the objective.

- A practical decoding algorithm that introduces minimal overhead compared to standard controlled decoding methods.

- Extensive experimental evaluation.

Concerns raised by reviewers

- The assumption that value functions are available for all objectives.

- Reliance on pretrained reward models.

- Comparisons to robust alignment baselines.

- Motivation of the problem setup.

During the rebuttal, the authors provide additional experimental results and clarifications that address most of these concerns. Overall, this paper makes a sound contribution to multi-objective inference-time alignment.

**Reviewer Concerns:**

I think all concerns are addressed during the rebuttal:

- The assumption that value functions are available for all objectives.

- Reliance on pretrained reward models.

- Comparisons to robust alignment baselines.

- Motivation of the problem setup.

**Reviewer Scores:**

Will likely increase to accept

---

### Decision · Program_Chairs · 2026-01-26

Accept (Poster)